# Research

inorganic chemistry

precursor speciation, growth of scorodite, ferrihydrite, poorly crystalline ferric arsenate, atmospheric scorodite synthesis

**Authors for correspondence:**
Xincun Tang
e-mail: tangxincun@csu.edu.cn
Yang Wang
e-mail: swsy250@csu.edu.cn

# The effect of precursor speciation on the growth of scorodite in an atmospheric scorodite synthesis

Zhihao Rong, Xincun Tang, Liping Wu, Xi Chen, Wei Dang, Xing Li, Liuchun Huang and Yang Wang

College of Chemistry and Chemical Engineering, Central South University, Changsha 410083, People's Republic of China

ZR, 0000-0002-1877-6769; YW, 0000-0002-3683-7633

In this study, we propose a growth pathway of scorodite in an atmospheric scorodite synthesis. Scorodite is a non-direct product, which is derived from the transformation of its precursor. Different precursor speciation leads to different crystallinity and morphology of synthesized scorodite. At 10 and $20 \, g \, l^{-1}$ initial arsenic concentration, the precursor of scorodite is identified as ferrihydrite. At $10 \, g \, l^{-1}$ initial arsenic concentration, low arsenic concentration is unfavourable to the complex between arsenate and ferrihydrite, inhibiting the transformation of ferrihydrite into scorodite. The synthesized scorodite is 1–3 µm in size. At $20 \, g \, l^{-1}$ initial arsenic concentration, higher arsenic concentration favours the complex between arsenate and ferrihydrite. The transformation process is accessible. Large scorodite in the particle size of 5–20 µm with excellent crystallinity is obtained. However, the increasing initial arsenic concentration is not always a positive force for the growth of scorodite. When initial arsenic concentration increases to $30 \, g \, l^{-1}$, $Fe(O,OH)_6$ octahedron preferentially connects to $As(O,OH)_4$ tetrahedron to form $FeH_2AsO_4^{2+}$ or $FeHAsO_4^+$ ion. Fe–As complex ions accumulate in solution. Once the supersaturation exceeds the critical value, the Fe–As complex ions deprotonate and form poorly crystalline ferric arsenate. Even poorly crystalline ferric arsenate can also transform to crystalline scorodite, its transformation process is much slower than ferrihydrite. Therefore, incomplete developed scorodite with poor crystallinity is obtained.

# 1. Introduction

Arsenic is a typical carcinogenic element, which has been identified as a group 1 carcinogen by International Agency for

Research on Cancer (IARC) [1]. It is often accompanied with non-ferrous metal (e.g. Ni, Sn, Pb, Cu and Zn) [2,3]. The occurrence of arsenic release during non-ferrous metal metallurgy poses a huge threat to the environment and human health all around the world [4]. The toxicity of arsenic links to its solubility. Accordingly, the immobilization of arsenic is of great concern. A good carrier for arsenic can restrain the release of arsenic, protecting humans from exposure to arsenic. Crystalline scorodite has been considered to be one of the most suitable arsenic immobilization materials because of its low solubility, high arsenic content, high filterability and low water content of precipitates [5,6]. The synthesis of scorodite can be carried out under both hydrothermal and atmospheric conditions [7–10]. However, the application of hydrothermal scorodite synthesis is inhibited by its high investment in equipment and large energy consumption [9]. The atmospheric scorodite synthesis by injecting oxygen to a Fe(II)–As(V) mixed solution has become a popular method due to its easy operation, high efficiency for arsenic precipitation and stable products [9,10]. Stability is the most important indicator to evaluate an immobilization material. Large and well-crystalline scorodite is preferable due to its good stability [11]. Some researchers have focused on the effect of different reaction conditions on the growth of scorodite [12–15]. However, to the best of our knowledge, the mechanism analysis of scorodite growth in this process has not been investigated.

The synthesis of scorodite is commonly recognized as a non-direct process. The formation and growth of scorodite occur via the transformation of its precursor. Ferrihydrite and poorly crystalline ferric arsenate were the most common precursors of scorodite, which were observed in other scorodite synthesis process [16,17]. Few reports pointed out the formation of X-ray amorphous precursor during atmospheric scorodite synthesis [18]. However, further characterization has not been carried out. In addition, the particle size of scorodite is influenced by its precursor. Larger scorodite is synthesized when the size of the precursor is increased by agglomeration [19–21]. Even though the precursor plays a significant role in the growth of scorodite during atmospheric scorodite synthesis, the speciation of the precursor has not been characterized. Moreover, the relationship between the growth of scorodite and the speciation of the precursor has not been established.

In this study, scorodite was synthesized by injecting pure oxygen into a Fe(II)–As(V) mixed solution at atmospheric pressure, reaction temperature of 95°C and pH of 1.0. The experiments were carried out at different initial arsenic concentrations (i.e. 10, 20 and 30 g $l^{-1}$) and different reaction times (i.e. 1, 3, 6 and 9 h). XRD, SEM and XPS were used to characterize the crystallinity and morphology of the synthesized samples. TCLP leaching test was used to directly evaluate the stability of synthesized samples. TG-DSC analysis was used to verify the formation of precursor during the synthesis. The precursor speciation was analysed by FTIR. UV–Vis spectrometer was used to characterize the solutions obtained at different reaction conditions.

# 2. Experimental

## 2.1. Materials

All chemicals used in experiments were analytical grade unless specified. High-purity arsenic trioxide (greater than or equal to 99.0%, Hunan Jinrun Tellurium Industry Co., Ltd, Hunan, China) was used as arsenic source. Ferrous sulfate ($FeSO_4 \cdot 7H_2O$, Sinopharm Chemical Reagent Co., Ltd, Shanghai, China) was used as iron source. Hydrogen peroxide was used as oxidant ($H_2O_2$, 30%, Sinopharm Chemical Reagent Co., Ltd, Shanghai, China). Sulfuric acid ($H_2SO_4$, 98%, Chengdu Chron Chemicals Co., Ltd, Chengdu, China) and sodium hydroxide (NaOH, Sinopharm Chemical Reagent Co., Ltd, Shanghai, China) were used for pH adjustment.

## 2.2. Preparation of As(V) stock solution

All solutions were prepared by deionized water. A measured amount of high-purity arsenic trioxide was dissolved into a sodium hydroxide solution to yield a Na/As molar ratio of 1. Then sufficient $H_2O_2$ solution was introduced dropwise into solution for oxidizing of arsenite to arsenate. The excess $H_2O_2$ was removed by heating. Then the prepared sodium dihydrogen arsenate stock solution was diluted with deionized water. An amount of $40 \, g \, l^{-1}$ As(V) stock solution was prepared and used for following experiments.

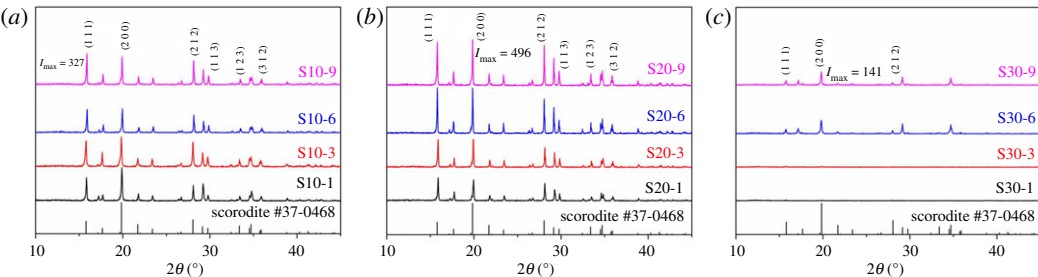

**Figure 1.** XRD patterns of samples synthesized at 10 (*a*), 20 (*b*) and 30 (*c*) g l$^{-1}$ initial arsenic concentration.

## 2.3. Experimental procedure

First, arsenic stock solution was transferred into a three-necked flask and diluted to desired arsenic concentration (10–30 g l$^{-1}$). The pH of solution was adjusted by $H_2SO_4$ to reach a pH value of 1.0. Then a measured amount of ferrous sulfate was added into arsenic solution to yield a Fe/As molar ratio of 1.5. The three-necked flask was placed into an oil bath and heated to 95°C. Thereafter, oxidation was performed by bubbling pure oxygen into solution. The solution was vigorously stirred by a magnetic stir bar during the synthesis. The oxygen flow rate was fixed at 0.5 l min$^{-1}$, while stirring rate was 1000 r.p.m. After synthesis, the precipitates were sampled by filtering reaction slurry immediately, and the filtrate residues were rinsed thoroughly by distilled water and dried at 60°C for 16 h. The samples will hereafter be referred to as S10-1, S10-3, S10-6, S10-9, S20-1, S20-3, S20-6, S20-9, S30-1, S30-3, S30-6, S30-9, corresponding the 10, 20, 30 g l$^{-1}$ initial arsenic concentration and 1, 3, 6, 9 h reaction time used for atmospheric scorodite synthesis, respectively.

## 2.4. Analysis and characterization

The X-ray diffraction (XRD) patterns were collected using a Bruker D8 diffraction instrument with Cu K$\alpha$ radiation (40 kV, 40 mA) (D8, Bruker, Germany). The samples were scanned from 10 to 45° 2$\theta$ with a step-size of 0.02° 2$\theta$. The morphology of samples was imaged on a scanning electron microscope (SEM, QUANTA FEG 250, FEI, USA). The spectra of O 1s peak were analysed using X-Ray photoelectron spectroscopy (XPS, ESCALAB 250Xi, Thermo Fisher Scientific, USA) with a monochromatic Al K$\alpha$ X-ray source. The step size of 0.05 eV was used for all scans. The binding energies were corrected using C 1s peak at 284.8 eV. The XPS spectra were fit using Thermo Avantage. The peak's full width at half maximum was fixed during the fitting. The leaching stability of samples was evaluated by a toxicity characteristic leaching procedure (TCLP) method. All TCLP tests were carried out according to standard 1311 of US-EPA by using a HAc–NaAc buffer solution of 4.93 pH. The leached arsenic concentration was determined using a PerkinElmer Optima 5300DV ICP-OES. The thermal analysis (TG-DSC) of synthesized samples was carried out on a Netzsch instrument (STA 449 F3, Netzsch, Germany) with 10 K min$^{-1}$ heating rate in air from room temperature up to 300°C. The Fourier transformed infrared (FTIR) spectra were recorded (Nicolet iS50, Thermo Fisher Scientific, USA). The KBr pellets (sample/KBr = 1 wt%) were prepared by mixing 2 mg samples and 200 mg KBr and characterized using transmission mode. The samples were scanned from 400 to 1000 cm$^{-1}$ with a resolution of 2 cm$^{-1}$. The solutions obtained at different reaction conditions were measured by a double-beam ultraviolet-visible spectrometer (TU-1901, Beijing Puxi Co., Ltd, Beijing, China). The solutions were diluted 15 times with dilute sulfuric acid (pH = 1.0) before measurement.

# 3. Results and discussion

## 3.1. Characterization of synthesized samples

Stability is the most important indicator to evaluate the quality of scorodite. Scorodite with large particle size and excellent crystallinity is preferable due to its strong stability. Therefore, XRD and SEM were used to characterize the synthesized samples. XRD patterns of samples synthesized at 10–30 g l$^{-1}$ initial arsenic concentrations are illustrated in figure 1*a*–*c*, respectively. The XRD patterns show that scorodite was successfully synthesized at 10–30 g l$^{-1}$ initial arsenic concentration. At different initial

10 g l$^{-1}$    20 g l$^{-1}$    30 g l$^{-1}$

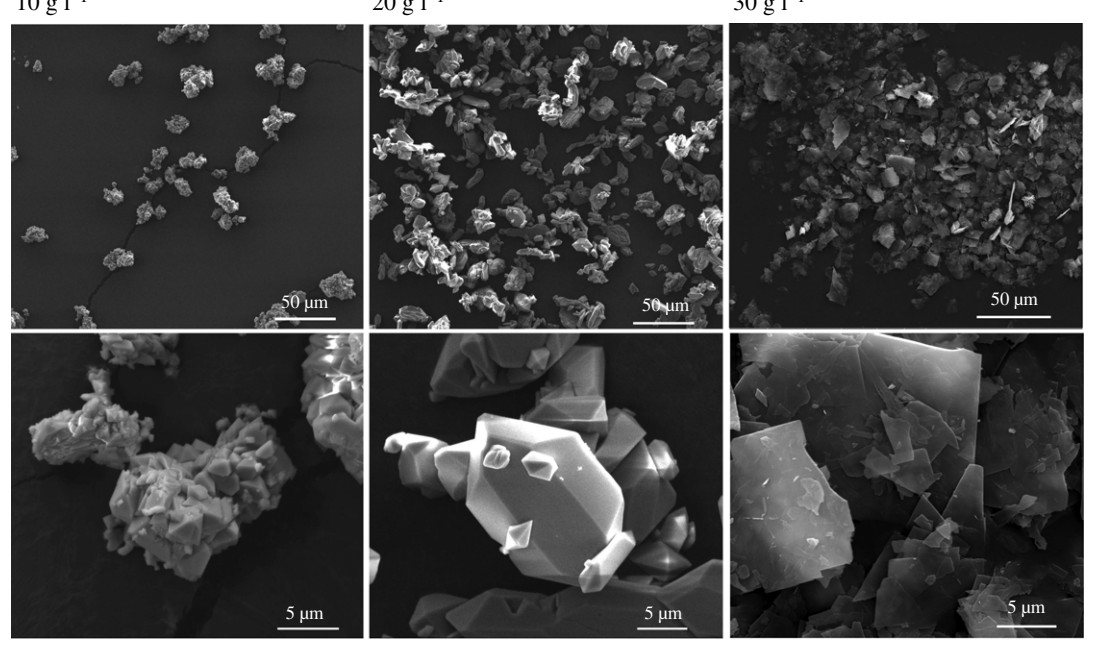

**Figure 2.** SEM images of samples synthesized at 10, 20 and 30 g l$^{-1}$ with 9 h reaction time.

arsenic concentration, the crystallization process and the crystallinity of products (S10-9, S20-9 and S30-9) are different. At 10 and 20 g l$^{-1}$ initial arsenic concentration, the crystallization of scorodite occurs as early as 1 h. The peaks of S10-9 and S20-9 match closely with PDF card of scorodite (JPSD #37-0468), which indicates the products have excellent crystallinity. However, when initial arsenic concentration increases to 30 g l$^{-1}$, the crystallization process is different from those at 10 and 20 g l$^{-1}$. The crystallization of scorodite occurs only when reaction time increases to more than 6 h. Even though S30-6 and S30-9 are identified as scorodite, their crystallinity is poorer than those synthesized at 10 and 20 g l$^{-1}$ initial arsenic concentration. The peaks at 15.8°, 19.8° and 28.0° are the main peaks of scorodite, corresponding to (1 1 1), (2 0 0) and (2 1 2) crystal plane, respectively. However, the intensity of these peaks is weak. The peaks located at 29.8°, 33.4° and 36.0° are absent, suggesting the growth of (1 1 3), (1 2 3) and (3 1 2) crystal planes are inhibited. In summary, at 30 g l$^{-1}$ initial arsenic concentration, the growth of scorodite is inhibited.

Figure 2 shows the SEM images for S10-9, S20-9 and S30-9. The S10-9 sample appears to be highly aggregated, consisting of primary scorodite particles roughly 1–3 µm in size. The principle of minimum surface energy is the reason why aggregation occurs. Small particles possess higher surface energy and tend to aggregate to reduce it. The distribution of particle size is not uniform in the image of S20-9. The large particles are approximately 20 µm in size, while the small particles are approximately 5 µm in size. In general, larger particles are obtained in the case of 20 g l$^{-1}$ initial arsenic concentration. The morphology changes significantly when initial arsenic concentration increases to 30 g l$^{-1}$. The SEM image of S30-9 shows that the sample has lamellar structure. The result is consistent with XRD characterization that the growth of some crystal planes is inhibited. The scorodite with lamellar structure was also observed in a previous study, which was identified as incomplete developed scorodite [22].

XPS was further used to characterize the surface of synthesized samples. XPS spectra are illustrated in figure 3a–i. The O 1s peaks are deconvoluted into two or three peaks. The peaks located at 532.7 eV and 531.4–531.5 eV correspond to surface adsorbed water and surface hydroxyl, respectively [23,24]. The peak at 530.9 eV is attributed to surface lattice oxygen, namely, oxygen in scorodite. At 10 g l$^{-1}$ initial arsenic concentration, no surface lattice oxygen is observed even when reaction time increases to 6 h. It can be seen from figure 3c, for S10-9, a new peak emerges at 530.9 eV, representing the occurrence of crystallization. At 20 g l$^{-1}$ initial arsenic concentration, surface lattice oxygen appears when reaction time is 3 h, which is earlier than the 9 h in 10 g l$^{-1}$. As shown in figure 3d–f, the fraction of surface lattice oxygen is 18.7% for S20-3, whereas 55.6%, 66.2% for S20-6 and S20-9, respectively. At 30 g l$^{-1}$ initial arsenic concentration, the peak of lattice oxygen appears only when reaction time increases to 9 h. Figure 3i illustrates that the fraction of surface lattice oxygen of S30-9 is as low as 20.7%. XPS

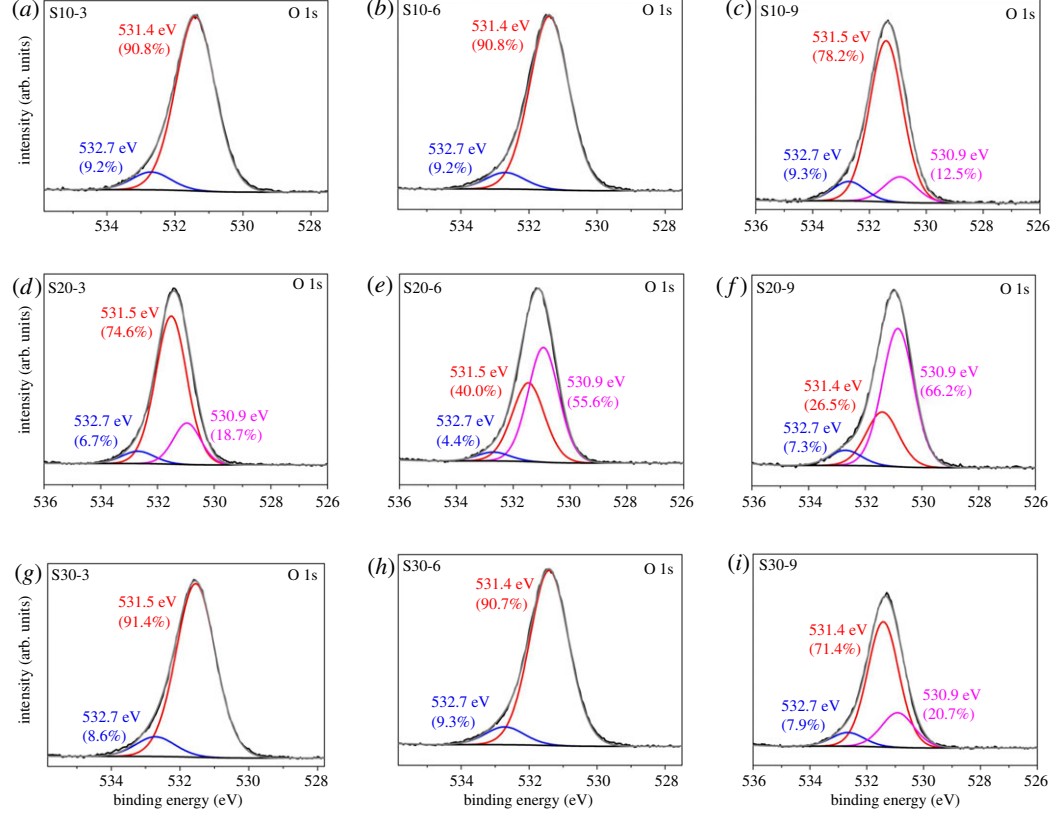

**Figure 3.** XPS O 1s spectra of S10-3 (*a*), S10-6 (*b*), S10-9 (*c*), S20-3 (*d*), S20-6 (*e*), S20-9 (*f*), S30-3 (*g*), S30-6 (*h*), S30-9 (*i*).

**Table 1.** TCLP leaching results.

| samples | arsenic leaching concentrations (mg l$^{-1}$) |
| --- | --- |
| S10-1 | 42.01 |
| S10-3 | 19.08 |
| S10-6 | 1.47 |
| S10-9 | 1.27 |
| S20-1 | 7.55 |
| S20-3 | 0.33 |
| S20-6 | 0.18 |
| S20-9 | <0.01 |
| S30-1 | 707.30 |
| S30-3 | 1896.17 |
| S30-6 | 8.99 |
| S30-9 | 3.40 |

analysis suggests that $20 \, g \, l^{-1}$ initial arsenic concentration is more suitable for the crystallization of scorodite compared with 10 and $30 \, g \, l^{-1}$. This result is justified by faster crystallization process and higher content of surface lattice oxygen.

TCLP leaching test was carried out to directly evaluate the stability of the synthesized samples. Table 1 shows that the samples synthesized at $20 \, g \, l^{-1}$ initial arsenic concentration exhibit the lowest arsenic leaching concentrations compared with those synthesized at 10 and $30 \, g \, l^{-1}$. Extremely, for S20-9, the leached arsenic concentration in HAc–NaAc leaching fluid is lower than $0.01 \, mg \, l^{-1}$, which is much lower than the permitting value of $5 \, mg \, l^{-1}$. This result is consistent with XRD, SEM and XPS characterizations. Large and well-crystalline scorodite exhibits strong stability.

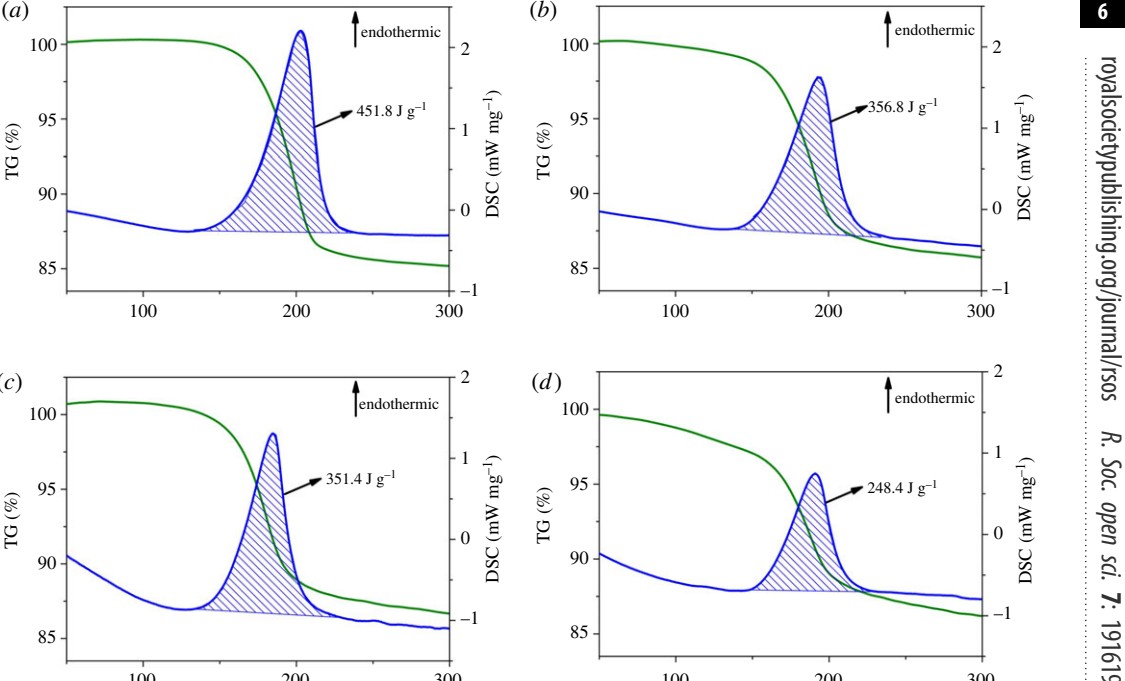

**Figure 4.** TG-DSC analysis of S20-9 (*a*), S10-1 (*b*), S20-1 (*c*) and S30-1 (*d*).

The effect of initial arsenic concentration on the growth of scorodite has been elucidated. However, the explanation for these phenomena is absent. Scorodite is a non-direct product, which is derived from the transformation of its precursor. The characterization of precursor speciation may help us to better understand the growth of scorodite.

## 3.2. Characterization of the precursor of scorodite

Figure 4*a–d* shows TG and DSC curves of synthesized samples. TG curves exhibit a weight loss stair in the range of 160–220°C. Meantime, DSC curves show an endothermic peak in similar range. These results suggest the presence of crystal water in synthesized samples. Scorodite contains a large amount of crystal water. The fraction of crystal water in amorphous or poorly crystalline structure is small. Therefore, the endothermic enthalpy of crystal water decomposition can be used as an indicator to identify the possible formation of ferrihydrite or poorly crystalline ferric arsenate. S20-9 was used as a reference material due to its excellent crystallinity. The DSC curve of S20-9 is shown in figure 4*a*, the endothermic enthalpy of crystal water decomposition is 451.8 J g$^{-1}$ for S20-9. As shown in figure 4*b–d*, the value is 356.8 J g$^{-1}$ for S10-1, while 351.4 and 248.8 J g$^{-1}$ for S20-1 and S30-1, respectively. The values are much lower than that of S20-9, suggesting the presence of amorphous or poorly crystalline precursor in synthesized samples.

The growth of scorodite is dependent on the speciation of its precursor. FTIR was further used for characterizing the speciation of precursor in synthesized samples. The displays of FTIR spectra in the scanning range of 400–1000 cm$^{-1}$ are illustrated in figure 5*a–c*. As shown in figure 5*a,b*, the band at 469–478 cm$^{-1}$ is observed in all spectra, which is attributed to the deformation of hydroxyl group in ferrihydrite [25]. Moreover, the band at 581–585 cm$^{-1}$ is also characteristic of ferrihydrite [26]. At 10 and 20 g l$^{-1}$ initial arsenic concentration, ferrihydrite is successfully identified as the precursor of scorodite in this process. The bands at 816 and 862 cm$^{-1}$ are attributed to the As–O stretching vibration. It is well established that arsenate can complex with ferrihydrite to form a bidentate binuclear surface complex of arsenate–ferrihydrite [27,28]. In the case of bidentate binuclear complexation, two of the four As–O bonding structures are complexed to ferrihydrite (As–O–Fe), but the other two are present as uncomplexed As–O. The band at 862 cm$^{-1}$ is attributed to the uncomplexed As–O band in arsenate–ferrihydrite, while the band at 816–823 cm$^{-1}$ is assigned to complexed As–O band (As–O–Fe) [29]. As shown in figure 5*a*, only a single As–O band is observed in the spectra, indicating the complex between arsenate and ferrihydrite is weak in the case of 10 g l$^{-1}$

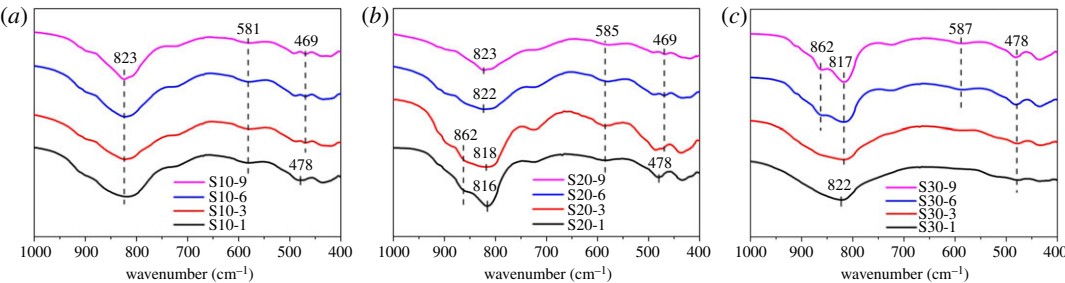

**Figure 5.** FTIR spectra of samples synthesized at 10 (*a*), 20 (*b*) and 30 (*c*) g l$^{-1}$ initial arsenic concentration.

initial arsenic concentration. It can be seen from figure 5*b* that the splitting of single As–O band into two bands is observed in the spectra of S20-1 and S20-3. This result suggests the formation of the bidentate binuclear complex of arsenate–ferrihydrite. Higher initial arsenic concentration is favourable to the complex between arsenate and ferrihydrite. As reaction time increases from 1 to 9 h, the As–O band at 862 cm$^{-1}$ is fading out. At the same time, another As–O band shifts gradually from 816 to 823 cm$^{-1}$, representing that the complex between arsenate and ferrihydrite has been weakened [29,30]. This is because the precipitation of scorodite decreases arsenic concentration. As shown in figure 5*c*, at 30 g l$^{-1}$ initial arsenic concentration, the band at 478 cm$^{-1}$ is weak while the band at 587 cm$^{-1}$ is almost invisible in the spectra of S30-1 and S30-3. The results indicate the formation of ferrihydrite is inhibited in the presence of high arsenic concentration. The As–O bands at 822 and 817 cm$^{-1}$ in the spectra of S30-1 and S30-3 are attributed to complexed As–O band (As–O–Fe). The characteristic band of ferrihydrite is weak. The splitting As–O band, which is characteristic of arsenate–ferrihydrite complex, is not observed. Crystalline scorodite is absent which is identified by XRD. Therefore, the complexed As–O bands (As–O–Fe) are attributed to poorly crystalline ferric arsenate. The bands at 478 and 587 cm$^{-1}$ gradually become evident with increasing reaction time, which represents the increasing amount of ferrihydrite. The precursor gradually turns from poorly crystalline ferric arsenate into ferrihydrite with increasing reaction time. As reaction time increases from 1 to 9 h, a new uncomplexed As–O band emerges at 862 cm$^{-1}$. The result indicates the complex between arsenate and ferrihydrite is present for S30-6 and S30-9.

UV spectra are shown in figure 6*a,b*. Figure 6*a* illustrates the spectra of solutions obtained at different initial arsenic concentrations. The peak at 290 nm is assigned to $FeH_2AsO_4^{2+}$ or $FeHAsO_4^{+}$ ion [31]. The absorbance at 290 nm increases significantly when initial arsenic concentration increases to 30 g l$^{-1}$. This result indicates that the complex between $Fe(O,OH)_6$ octahedron and $As(O,OH)_4$ tetrahedron is more likely to occur at 30 g l$^{-1}$ initial arsenic concentration. Higher initial arsenic concentration is favourable to the formation of $FeH_2AsO_4^{2+}$ or $FeHAsO_4^{+}$ ions. Subsequently, the deprotonation of Fe–As complex ions leads to the formation of poorly crystalline ferric arsenate [32]. As shown in figure 6*b*, the absorbance at 290 nm decreases gradually with increasing reaction time. The precipitation of arsenic decreases arsenic concentration, weakening the complex between $Fe(O,OH)_6$ octahedron and $As(O,OH)_4$ tetrahedron. This is the reason why the precursor speciation turns from poorly crystalline ferric arsenate to ferrihydrite with increasing reaction time.

## 3.3. The effect of precursor speciation on the growth of scorodite

In many systems, the crystallization process is indirect [33]. Poorly ordered precursor forms first, and then transforms to crystalline phase. This phenomenon can be explained by the Ostwald step rule [33]. Less stable phase forms first because the requirement for interfacial energy is less stringent [34]. Then, according to the minimum energy principle, the precursor phase tends to crystallize to reduce its energy [35]. This rule also applies to the formation of crystalline scorodite. The precursor forms first, and then transforms to scorodite. Different initial arsenic concentration leads to different precursor speciation, and consequently leads to different crystallinity and morphology of synthesized scorodite.

At 10 and 20 g l$^{-1}$ initial arsenic concentration, the oxidation of ferrous ions produces ferrihydrite. The transformation of ferrihydrite into scorodite goes circularly through four stages: (i) surface complex of arsenate and ferrihydrite, (ii) release of ferric ion by the dissolution of ferrihydrite, (iii) the adsorption of ferric ion on adsorbed arsenate, and (iv) re-adsorption of arsenate on adsorbed ferric ion [29,30]. This cycle ensures the transformation of ferrihydrite into scorodite. The inhibition of any

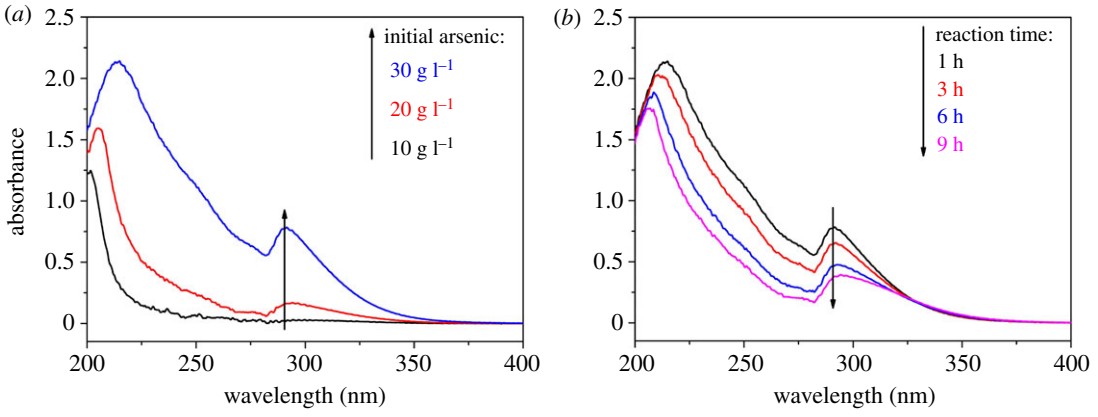

**Figure 6.** UV spectra of solutions obtained at different initial arsenic concentrations with reaction times of 1 h (*a*) and at different reaction times with initial arsenic concentration of 30 g l$^{-1}$ (*b*).

step in the process will constrain the growth of scorodite. At 10 g l$^{-1}$ initial arsenic concentration, low arsenic concentration inhibits the complex between arsenate and ferrihydrite. The transformation process is inhibited. Small scorodite with a particle size of 1–3 µm is obtained. At 20 g l$^{-1}$ initial arsenic concentration, higher arsenic concentration favours the complex between arsenate and ferrihydrite. The transformation pathway is accessible. Therefore, large scorodite with excellent crystallinity is obtained. However, the increasing arsenic concentration is not always a positive force for scorodite synthesis. At 30 g l$^{-1}$ initial arsenic concentration, too high arsenic concentration inhibits the formation of ferrihydrite. Fe(O,OH)$_6$ octahedron tends to connect with As(O,OH)$_4$ tetrahedron to form FeH$_2$AsO$_4$$^{2+}$ or FeHAsO$_4$$^+$ ion. The Fe–As complex ions accumulate in solution. Once the supersaturation exceeds a critical value, the deprotonation of FeH$_2$AsO$_4$$^{2+}$ or FeHAsO$_4$$^+$ ions occurs and produces poorly crystalline ferric arsenate. The transformation of poorly crystalline ferric arsenate into scorodite has been investigated in previous studies. Some authors pointed out that the transformation process was an aggregation-based growth. The crystallization of scorodite occurred within the aggregated precursor. The process appeared to be an auto-catalytic reaction. An induction period was required at the beginning for the accumulation of nuclei [16]. Other authors proposed that the phase transformation proceeds via a cycle of dissolution–reprecipitation, which is similar to that of ferrihydrite [17]. The mechanism of the transformation process is still a matter of debate. However, in any case, the transformation process of poorly crystalline ferric arsenate is much slower than that of ferrihydrite. The induction period or low solubility of poorly crystalline ferric arsenate inhibits the transformation process. Crystalline scorodite appears only when reaction time increases to more than 6 h. With the decreasing arsenic concentration, ferrihydrite gradually forms and then arsenate complexes to ferrihydrite, which accelerate the crystallization process. However, the growth of scorodite has already been inhibited by the slow transformation process at the beginning. Therefore, incomplete developed scorodite with poor crystallinity is obtained.

## 4. Conclusion

In this study, the precursor speciation at different initial arsenic concentration was analysed. Scorodite grows via the transformation of its precursor. At 10 and 20 g l$^{-1}$ initial arsenic concentration, ferrihydrite is identified as the precursor of scorodite. At 10 g l$^{-1}$ initial arsenic concentration, low arsenic concentration inhibits the complex between arsenate and ferrihydrite. The transformation process is inhibited. Therefore, scorodite with small particle size is obtained. At 20 g l$^{-1}$ initial arsenic concentration, higher initial arsenic concentration favours the complex between arsenate and ferrihydrite. The transformation pathway is accessible. Large scorodite with excellent crystallinity is obtained. At 30 g l$^{-1}$ initial arsenic concentration, Fe(O,OH)$_6$ octahedron preferentially connects to As(O,OH)$_4$ tetrahedron. The Fe–As complex ions accumulate in solution. Once the supersaturation exceeds the critical value, the deprotonation of FeH$_2$AsO$_4^{2+}$ or FeHAsO$_4^+$ ions occurs and produces poorly crystalline ferric arsenate. The transformation process of poorly crystalline ferric arsenate is much slower than that of ferrihydrite. Due to the lack of ferrihydrite and the slow transformation process of poorly crystalline ferric arsenate, the growth of scorodite is inhibited in the case of 30 g l$^{-1}$ initial arsenic concentration. Incomplete developed scorodite with poor crystallinity is obtained.

Data accessibility. Data have been deposited into Dryad Digital Repository: https://doi.org/10.5061/dryad.j3tx95x89 [36].
Authors' contributions. Z.R. carried out the experiment, data analysis and sequence alignments, participated in the design of the study and drafted the manuscript; X.T., L.W., X.C., W.D., L.H. and X.L. carried out the statistical analyses; X.T. and Y.W. conceived of the study, designed the study, coordinated the study and helped draft the manuscript. All authors gave final approval for publication.
Competing interests. We declare we have no competing interest.
Funding. This work was supported by National Natural Science Foundation of China (grant no. 21476268).

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
