## [Reviewer comments · Royal Society Open Science]

Review History

RSOS-191619.R0 (Original submission)

Review form: Reviewer 1

Is the manuscript scientifically sound in its present form?

Yes

Are the interpretations and conclusions justified by the results?

Yes

Is the language acceptable?

Yes

Do you have any ethical concerns with this paper?

No

Have you any concerns about statistical analyses in this paper?

No

Recommendation?

Accept with minor revision (please list in comments)

Comments to the Author(s)

Dear authors,

The manuscript presents an interesting results. But a deep discussion about crystal growth should be considered. Once that is the main object in the manuscript.

It was proposed that higher concentration, the growth was reduced while a kind of complex Fe (O, OH) 6 / As (O, OH) 4 was formed. But there isn't any characterization about this complex in higher concentration. Maybe it could be identified by UV-Vis?

Review form: Reviewer 2

Is the manuscript scientifically sound in its present form?

Yes

Are the interpretations and conclusions justified by the results?

Yes

Is the language acceptable?

Yes

Do you have any ethical concerns with this paper?

No

Have you any concerns about statistical analyses in this paper?

No

Recommendation?

Accept with minor revision (please list in comments)

Comments to the Author(s)

The authors investigated the effect of precursor speciation on the growth of scorodite at atmospheric pressure. Overall, there are some minor flaws and the authors should address the following issues before it can possibly be published.

P1 Abstract "Scorodite is a non-direct production."

I think the authors meant "product".

P2 2.2 Experimental procedure "...followed by the introduction of pure oxygen..."

How was the oxygen introduced? By bubbling through the reaction solution or not?

Please also include the approximate flow rate of oxygen.

P2 2.3 Analysis and characterization "The FT-IR spectra were obtained on a Thermo Fisher Scientific, Nicolet IS 50 spectrometer ranging from 400 to 1000 cm⁻¹."

Please clarify whether transmission mode or ATR mode was used. If transmission mode was used, please include the matrix material (e.g. KBr)

Decision letter (RSOS-191619.R0)

05-Nov-2019

Dear Dr Tang:

Title: The effect of precursor speciation on the growth of scorodite in an atmospheric scorodite synthesis

Manuscript ID: RSOS-191619

The editor assigned to your manuscript has now received comments from reviewers. We would like you to revise your paper in accordance with the referee and Subject Editor suggestions which can be found below (not including confidential reports to the Editor). Please note this decision does not guarantee eventual acceptance.

Please submit your revised paper before 28-Nov-2019. Please note that the revision deadline will expire at 00.00am on this date. If we do not hear from you within this time then it will be assumed that the paper has been withdrawn. In exceptional circumstances, extensions may be possible if agreed with the Editorial Office in advance. We do not allow multiple rounds of revision so we urge you to make every effort to fully address all of the comments at this stage. If deemed necessary by the Editors, your manuscript will be sent back to one or more of the original reviewers for assessment. If the original reviewers are not available we may invite new reviewers.

Please also include the following statements alongside the other end statements. As we cannot publish your manuscript without these end statements included, if you feel that a given heading is not relevant to your paper, please nevertheless include the heading and explicitly state that it is not relevant to your work.

- Acknowledgements

- Funding statement

Please include a funding section after your main text which lists the source of funding for each author.

RSC Associate Editor:
Comments to the Author:
(There are no comments.)

RSC Subject Editor:
Comments to the Author:
(There are no comments.)

Reviewers' Comments to Author:
Reviewer: 1

Comments to the Author(s)
Dear authors,

The manuscript presents an interesting results. But a deep discussion about crystal growth should be considered. Once that is the main object in the manuscript.

It was proposed that higher concentration, the growth was reduced while a kind of complex Fe (O, OH) 6 / As (O, OH) 4 was formed. But there isn't any characterization about this complex in higher concentration. Maybe it could be identified by UV-Vis?

Reviewer: 2

Comments to the Author(s)
The authors investigated the effect of precursor speciation on the growth of scorodite at atmospheric pressure. Overall, there are some minor flaws and the authors should address the following issues before it can possibly be published.

P1 Abstract "Scorodite is a non-direct production."
I think the authors meant "product".

P2 2.2 Experimental procedure "...followed by the introduction of pure oxygen..."
How was the oxygen introduced? By bubbling through the reaction solution or not?
Please also include the approximate flow rate of oxygen.

P2 2.3 Analysis and characterization “The FT-IR spectra were obtained on a Thermo Fisher Scientific, Nicolet IS 50 spectrometer ranging from 400 to 1000 cm⁻¹.”

Please clarify whether transmission mode or ATR mode was used. If transmission mode was used, please include the matrix material (e.g. KBr)

Author's Response to Decision Letter for (RSOS-191619.R0)

See Appendix A.

RSOS-191619.R1 (Revision)

Review form: Reviewer 2

Is the manuscript scientifically sound in its present form?

Yes

Are the interpretations and conclusions justified by the results?

Yes

Is the language acceptable?

Yes

Do you have any ethical concerns with this paper?

No

Have you any concerns about statistical analyses in this paper?

No

Recommendation?

Accept as is

Comments to the Author(s)

The authors addressed my questions and I appreciate it.

Decision letter (RSOS-191619.R1)

10-Dec-2019

Dear Dr Tang:

Title: The effect of precursor speciation on the growth of scorodite in an atmospheric scorodite synthesis

Manuscript ID: RSOS-191619.R1

It is a pleasure to accept your manuscript in its current form for publication in Royal Society Open Science. The chemistry content of Royal Society Open Science is published in collaboration with the Royal Society of Chemistry.

RSC Associate Editor:
Comments to the Author:
(There are no comments.)

RSC Subject Editor:
Comments to the Author:
(There are no comments.)

Reviewer(s)' Comments to Author:
Reviewer: 2

Comments to the Author(s)
The authors addressed my questions and I appreciate it.

Appendix A

Dear editor and reviewers,

Thank you for your letter and for the reviewers' comments concerning our manuscript entitled "The effect of precursor speciation on the growth of scorodite in an atmospheric scorodite synthesis"(Manuscript ID: RSOS-191619). The comments are all valuable and very helpful for our manuscript, which have provided the important guiding significance to our researches. We have studied the comments carefully and have made the corrections accordingly. Changes and additions are marked in blue in revision. The data in Dryad Digital Repository has been updated. The main corrections in the revision and the response to the reviewer's comments are shown as follows. We have tried our best to polish the manuscript and hope what we have done could meet your approval.

Thanks a lot.

Sincerely yours,

Xincun Tang

Reviewers' comments:

Reviewer: 1

The manuscript presents an interesting result.

1. But a deep discussion about crystal growth should be considered. Once that is the main object in the manuscript.

Reply: Thanks for the reviewer's kind comments. According to your advice, we have added the discussion about crystal growth in manuscript as follow:

"In many systems, the crystallization process is indirect³³. Poorly ordered precursor forms first, and then transforms to crystalline phase. This phenomenon can be explained by Ostwald step rule³³. Less stable phase forms first because the requirement for interfacial energy is less stringent³⁴. Then, according to the minimum energy principle, the precursor phase tends to crystallize to reduce its energy³⁵. This rule is also applies to the formation of crystalline scorodite. The precursor forms first, and then transforms to scorodite. Different initial arsenic concentration leads to different precursor speciation, and consequently leads to different crystallinity and morphology of synthesized scorodite."

33. P. R. ten Wolde and D. Frenkel, *PCCP Phys. Chem. Chem. Phys.*, 1999, **1**, 2191-2196.

34. G. P. Demopoulos, *Hydrometallurgy*, 2009, **96**, 199-214.

35. Y. Wang, Z. Rong, X. Tang, S. Cao, X. Chen, W. Dang and L. Wu, *Applied Surface Science*, 2019, **496**, 143719.

2. It was proposed that higher concentration, the growth was reduced while a kind of complex $\text{Fe}(\text{O}, \text{OH})_6 / \text{As}(\text{O}, \text{OH})_4$ was formed. But there isn't any characterization about this complex in higher concentration. Maybe it could be identified by UV-Vis?

Reply: Thanks for the reviewer's suggestions. UV-Vis is indeed a method to identify the Fe-As complex ion. According to your suggestion, solutions obtained at different reaction conditions were measured by a double-beam ultraviolet-visible spectrometer (TU-1901, Beijing Puxi Co., Ltd, Beijing, China). The solutions were diluted 15 times with dilute sulphuric acid (pH=1.0) before measurement. Results and analysis were written as follow in manuscript:

"UV spectra are shown in Fig. 6a and b. Fig. 6a illustrates the spectra of solutions obtained at different initial arsenic concentrations. The peak at 290 nm is assigned to $\text{FeH}_2\text{AsO}_4^{2+}$ or FeHAsO_4^+ ion³¹. The absorbance at 290 nm increases significantly when initial arsenic concentration increases to 30 g·L⁻¹. This result indicates that the complex between $\text{Fe}(\text{O}, \text{OH})_6$ octahedron and $\text{As}(\text{O}, \text{OH})_4$ tetrahedron is more likely to occur at 30 g·L⁻¹ initial arsenic concentration. Higher initial arsenic concentration is favourable to the formation of $\text{FeH}_2\text{AsO}_4^{2+}$ or FeHAsO_4^+ ions. Subsequently, the deprotonation of Fe-As complex ions leads to the formation of poorly crystalline ferric arsenate³². As shown in Fig. 6b, the absorbance at 290 nm decreases gradually with increasing reaction time. The precipitation of arsenic decreases arsenic concentration, weakening the complex between $\text{Fe}(\text{O}, \text{OH})_6$ octahedron and $\text{As}(\text{O}, \text{OH})_4$ tetrahedron. This is the reason why the precursor speciation turns from poorly crystalline ferric arsenate to ferrihydrite with increasing reaction time."

Fig.6 UV spectra of solutions obtained at different initial arsenic concentration with reaction time of 1 h (a) and at different reaction time with initial arsenic concentration of 30 g·L⁻¹ (b).

31. L. Chai, J. Yang, N. Zhang, P. J. Wu, Q. Li, Q. Wang, H. Liu and H. Yi, *Chemosphere*, 2017, **182**, 595-604.

32. J. Yang, L. Chai, M. Yue and Q. Li, *RSC Advances*, 2015, **5**, 103936-103942.

Reviewer: 2

The authors investigated the effect of precursor speciation on the growth of scorodite at atmospheric pressure. Overall, there are some minor flaws and the authors should address the following issues before it can possibly be published.

1. P1 Abstract “Scorodite is a non-direct production.”

I think the authors meant “product”.

Reply: Thanks for pointing out the error and we are very sorry for making such a mistake. We have corrected it in our revision.

2. P2 2.2 Experimentl procedure “...followed by the introduction of pure oxygen...”

How was the oxygen introduced? By bubbling through the reaction solution or not?

Please also include the approximate flow rate of oxygen.

Reply: Thanks for the reviewer’s carefully reading and we are sorry for this confusing description.

The oxidation was performed by bubbling pure oxygen into solution. The oxygen flow rate was fixed at 0.5 L·min⁻¹.

In addition, after reading your comments, we checked the experimental procedure section in our manuscript. We found that this section was not easy to follow. Therefore, we have rewritten the Experimentl procedure section in manuscript as follow:

“At first, arsenic stock solution was transferred into a three-necked flask and diluted to desired arsenic concentration (10-30 g·L⁻¹). The pH of solution was adjusted by H₂SO₄ to reach a pH value of 1.0. Then a measured amount of ferrous sulphate was added into arsenic solution to yield a Fe/As molar ratio of 1.5. The three-necked flask was placed into an oil bath and heated to 95°C Thereafter, oxidation was performed by bubbling pure oxygen into solution. The solution was vigorously stirred by a magnetic stir bar during the synthesis. The oxygen flow rate was fixed at 0.5 L·min⁻¹ while stirring rate was 1000 rpm.

After synthesis, the precipitates were sampled by filtering reaction slurry immediately, and the filtrate residues were rinsed thoroughly by distilled water and dried at 60 °C for 16 h. The samples will hereafter be referred to as S10-1, S10-3, S10-6, S10-9, S20-1, S20-3, S20-6, S20-9, S30-1, S30-3, S30-6, S30-9, corresponding to the 10, 20, 30 g·L⁻¹ initial arsenic concentration and 1, 3, 6, 9 h reaction time used for atmospheric scorodite synthesis, respectively.”

3. 2.3 Analysis and characterization “The FT-IR spectra were obtained on a Thermo Fisher Scientific, Nicolet IS 50 spectrometer ranging from 400 to 1000 cm⁻¹.”

Please clarify whether transmission mode or ATR mode was used. If transmission mode was used, please include the matrix material (e.g. KBr)

Reply: Thanks for the reviewer’s carefully reading and we are sorry for this confusing description. The FTIR spectra were obtained using transmission mode and KBr pellet technique. The expression about FTIR characterization was expressed in manuscript as follow:

“The Fourier Transformed Infrared spectra (FTIR) were recorded (Nicolet iS50, Thermo Fisher Scientific, USA). The KBr pellets (sample/KBr=1 wt%) were prepared by mixing 2 mg samples and 200 mg KBr and characterized using transmission mode. The samples were scanned from 400-1000 cm⁻¹ with a resolution of 2cm⁻¹.”

Our manuscript has been thoroughly revised according to suggestions of reviewers. Thank you for your time and your excellent question. We hope the manuscript can meet the requirements for the *Royal Society Open Science*. We are looking forward to obtaining your positive consideration.